# Medical Adhesives and Their Role in Laparoscopic Surgery—A Review of Literature

**DOI:** 10.3390/ma15155215

**Published:** 2022-07-28

**Authors:** Maciej Mazur, Wojciech Zakrzewski, Maria Szymonowicz, Zbigniew Rybak

**Affiliations:** 1Department of Surgery, 4th Military Teaching Hospital in Wroclaw, Weigla 5, 50-981 Wroclaw, Poland; lekmaciejmazur@gmail.com; 2Pre-Clinical Research Centre, Wroclaw Medical University, Bujwida 44, 50-345 Wroclaw, Poland; maria.szymonowicz@umw.edu.pl (M.S.); zbigniew.rybak@umw.edu.pl (Z.R.)

**Keywords:** medical adhesive, adhesive, fibrin, gelatin, stem cells

## Abstract

Laparoscopic surgery is undergoing rapid development. Replacing the traditional method of joining cut tissues with sutures or staples could greatly simplify and speed up laparoscopic procedures. This alternative could undoubtedly be adhesives. For decades, scientists have been working on a material to bond tissues together to create the best possible conditions for tissue regeneration. The results of research on tissue adhesives achieved over the past years show comparable treatment effects to traditional methods. Tissue adhesives are a good alternative to surgical sutures in wound closure. This article is a review of the most important groups of tissue adhesives including their properties and possible applications. Recent reports on the development of biological adhesives are also discussed.

## 1. Introduction

Adhesive is a generic term and includes other common terms such as adhesive, paste, gum, and bonding agent [1,2]. Adhesives are defined as nonmetallic substances capable of bonding materials together by joining surfaces. The bond itself has sufficient internal strength (cohesion). Already prehistoric hunters discovered that human or animal blood can adhere hair together and can be used as an adhesive. In ancient China, adhesives of plant origin were used, while the Egyptians used gum arabic, among other things, to thicken cosmetics. A spectacular breakthrough came with the development of the chemical industry in the early 20th century, which led to the discovery of phenolic, urea, epoxy, cyanoacrylate, and many other resins. Adhesives quickly found their way into medical applications [3].

Tissue adhesives are used to create functional connections between damaged tissues and between tissues and implanted biomaterials [4]. Millions of patients worldwide suffer from surgical wounds or incisions that require effective closure, accelerated healing, and tissue regeneration [5]. Surgical sutures and mechanical methods of tissue fusion are useful for conventional open surgery. Sutures and mechanical methods are not without risks of intolerance and toxicity [5,6]. An ideal tissue adhesive should meet the appropriate criteria to be widely used in clinical practice [7,8]:It should be biocompatible and biodegradableThe adhesive application should be as simple as possibleThe adhesive curing process should be relatively shortMechanical strength, cohesion, and adhesion in a humid environment should be appropriate for the specific applicationThe adhesive should not cause an inflammatory reaction with large swellingThe adhesive should be durable and stableProduction costs must be economically viable

Available tissue adhesives can be divided into natural and synthetic. Synthetic adhesives have strong mechanical properties, low degradation rates, and their degradation products are cytotoxic, which limits their clinical application [9]. Natural adhesives, including fibrin adhesives and collagen-based products, are effective in adhesion; unfortunately, their availability is limited due to autologous tissue isolation. In addition, researchers note the often poor tensile strength of adhesives [10]. When designing an adhesive, the aim is to optimize specific desired parameters while maintaining its overall characteristics [11]. Other important parameters for tissue adhesives are mechanical stability and biocompatible integration with other biomaterials [12,13]. An ideal tissue adhesive should form an interface that connects the biomaterial to the tissue. Ideal adhesives should be able to modulate cellular activity through active biological signals [14,15,16]. Recent research has focused on creating a hybrid system. This would consist of an adhesive that would have features of the cellular microenvironment to mimic the extracellular architecture and biology of a given tissue [12,17,18]. The following chapters of this manuscript focus on adhesives, their division into natural or artificial ones, and the following features of example materials. Additionally, the manuscript focuses on the very promising application of stem cells in regenerative medicine, especially when it comes to adhesives.

## 2. Natural Adhesives

### 2.1. Fibrin Adhesives

Fibrin adhesives are made from human or animal plasma, are biodegradable, easy to use, and can be stored at room temperature or in the refrigerator. Fibrin adhesive has strong hemostatic properties. The basic components of the adhesive are fibrinogen and thrombin [19,20]. In the presence of calcium chloride, thrombin breaks fibrinogen chains to form fibrin. After 2 weeks, the clot formed is completely absorbed; therefore, the adhesive does not cause an inflammatory response. Fibrin gels can serve as an autologous biodegradable scaffold. The tensile strength of a fibrin clot depends on the concentration of fibrinogen in the adhesive, which ranges from 40 to 115 mg/mL. The adhesive strength is much lower compared to cyanoacrylate adhesive. Fibrin adhesive has been used since the 1970s to achieve hemostasis in cardiac surgery [21,22] to seal vascular grafts and to treat aortic dissection in vascular surgery [23,24]. Prevention of sudden bleeding during surgery requires the support of fibrin adhesive with sutures, staples, or sponge because the strength of the adhesive is reduced in moist environments [25,26,27,28,29,30]. In neurosurgery, it can be used to close the dura to prevent leakage of cerebrospinal fluid (CSF) and to repair leaks in the dura [31,32,33]. The fibrin adhesive Tissucol has also found use in plastic surgery as a means to provide hemostasis in the treatment of burn wounds [34]. Adhesives can also be used in endoscopic procedures to treat peptic ulcers [35]. Fibrin adhesive can also be used in laparoscopic inguinal hernia repair using the Totally Extraperitoneal Repair (TEP) technique. Researchers compared the results of two groups of patients; in the first group, the polypropylene mesh was attached using staples, while in the second group it was attached using fibrin adhesive. The study showed a lower incidence of hematoma, pain, postoperative neuralgia, and a faster return to work in patients in the second group [36]. Another study conducted on patients treated laparoscopically with transabdominal preperitoneal mesh placement (TAPP—transabdominal preperitoneal repair) found no statistically significant difference in the patient’s postoperative hospital stay, recovery time, recurrence rate, and complications [37]. This allows to conclude that adhesive treatment leads to the same good treatment results as classical methods. One of the studies reports a case of a successful treatment of a large lymphangioma located in the mediastinum by drainage and fibrin adhesive injections [38]. Four-year follow-up showed no recurrence. Adhesive has been used to seal the pleura [39] and even to treat bronchial fistulas [40]. Pneumothorax is one of the complications in premature infants with respiratory failure and is associated with high mortality. In 12 cases of pleurodesis with fibrin adhesive, permanent pleural seal occurred [41]. One of the most common complications after kidney transplantation is lymphocele. The authors evaluated and compared three treatment procedures in these patients: open surgical drainage, laparoscopic drainage, and percutaneous drainage with fibrin adhesive injections. The use of the adhesive resulted in the best outcomes, with a lower recurrence rate of 7.7% compared to laparoscopic drainage at 54% and surgical drainage at 27.3%. In addition, the cost of treatment was lower, and the procedure could be performed on an outpatient basis under local anesthesia [42]. Furthermore, adhesive can be used in otolaryngology after radical lymph node excision to prevent lymphatic leakage [43]. The use of adhesives in knee replacement surgery has helped to reduce bleeding during surgery [44]. Research is ongoing into the use of fibrin adhesives as a carrier for anticancer drugs. Studies have been conducted on mice that were implanted with colon cancer cells into the subcutaneous tissue. The results of oxaliplatin used alone and in combination with fibrin adhesive were compared. Greater tumor apoptosis, decreased proliferation, and inhibition of tumor angiogenesis were observed in the group treated with oxaliplatin with fibrin adhesive [45]. The use of fibrin adhesive in the treatment of nerve injury is ongoing. A study used the sciatic nerve in mice [46]. It showed faster axonal regeneration, and better electrophysiological and functional outcome of the nerve after treatment with adhesive compared to microsurgical suturing. The use of adhesives also reduced the regeneration time of the injured nerve [47]. The main advantages of using fibrin-based adhesives are lack of toxicity, biocompatibility, and the natural degradation process. Fibrin-based adhesive can also be applied as foam [48]. It is worth noting that animal studies have proven the superiority of foam over the liquid form of the adhesive when in contact with blood [49]. Hybrid materials such as TachoComb [50], which is used in Europe and consists of a thin layer of lyophilized human fibrinogen and bovine thrombin placed on one side of a sheet of horse collagen, have proven their effectiveness in controlling bleeding [51,52]. The weaknesses of fibrin products include the method of manufacture, while the autologous isolation from tissue limits the availability of fibrin and makes fibrin adhesives relatively expensive. There are also some concerns about the safety of using thrombin from bovine sources, including allergic reactions and the transmission of infection. Horowitz and Busch [53] studied the safety of fibrin adhesives. In their study, the authors reported a risk of parvovirus transmission of approximately 1/500,000 vials of fibrinogen, whereas the risk of HIV, HCV, HBV, and HAV infection was 1 in 10 to the power of 15. The potential risk of Creutzfeldt–Jacob disease infection cannot be excluded either [54,55]. In addition, the adhesive shows low adhesion strength due to the poor cohesive properties, and may be difficult to apply because the polymerization time is difficult to control during application [56]. Bjork et al. [57] developed a new crosslinking method based on ruthenium and blue visible light that results in a three-fold increase in the mechanical strength and a ten-fold increase in the stiffness by adjusting the exposure time and culture duration on both fibrin- and collagen-based constructs.

### 2.2. Collagen Adhesives

Another group of natural adhesives are collagen-based adhesives. Collagen has adsorption properties of coagulation products that cause the adhesive to adhere to the wound by inducing platelet adhesion and activating clotting factors [58,59]. In the US, an adhesive called FloSeal [60] was registered to produce hemostasis in vascular surgery, while the Proceed product is designed to prevent and treat cerebrospinal fluid leaks [61]. Another novel collagen product is Costasis [62], which is a combination of autologous human plasma and a mixture of bovine collagen and thrombin. It has been used as a spray for surgical hemostasis and in endoscopic gastrointestinal bleeding [63]. However, researchers point out that, as with any zoonotic adhesive, there is a risk of transmission of infection and allergic reaction.

### 2.3. Gelatin-Based Adhesives

Gelatin adhesives have attracted the attention of researchers because of their many applications in soft tissue bonding. In addition, they can be used to prevent and treat both liquid and gaseous leaks [64]. Gelatin itself is a widely used collagen derivative which is obtained with use of a controlled process of hydrolysis, and is a major part of the skin, connective tissue, and skeletal system [65,66]. Adhesives based on this material are inexpensive to produce and their ingredients are widely available. Potential complications after adhesive use include vasoconstriction at the site of administration or movement of the adhesive substance from the site of administration to a distant site. Elvin et al. [67] demonstrated that photochemically crosslinked gelatin-based adhesive demonstrates a high adhesive strength (>100 kPa), a high elasticity (>600% elongation to break), and a high tensile strength (approximately 2.0 MPa). Some in vitro studies [68,69,70] have shown that calcium-independent microbial transglutaminase (mTG) can crosslink gelatin to form a gel within minutes. The mTG gelatin adhesive can bond to moist and wet tissues, and the adhesive strength is comparable or even better than the fibrin-based adhesive. Its surgical potential has been evaluated highly in in vivo studies using rats and pigs [71]. A gelatin–resorcinol adhesive crosslinked with formaldehyde (GRF) or glutaraldehyde (GRFG) is an interesting solution. The adhesive itself consists of the organic components resorcinol, gelatin, and water [72]. The curing process of the adhesive occurs in the presence of formaldehyde, glutaraldehyde, or under the influence of high temperature [73]. The cured bond is formed by heating a mixture in the ratio of three parts gelatin to one part resorcinol. Crosslinking occurs after approximately 30 s. Currently, there is one adhesive, BioGlue (CryoLife, Inc., 1655 Roberts Blvd NW, Kennesaw, GA 30144, USA) based on GRF that has been approved in the US by the FDA [74]. In BioGlue, the formaldehyde component was omitted due to its toxic potential. To date, no other GRF adhesive has been approved by the US FDA because the long-term results are not yet known. This adhesive has been used worldwide in a wide variety of surgical procedures including: cardiac surgery (aortic valve replacement, coronary artery bypass grafting, aortic dissection repair, and abdominal aortic aneurysm repair), as well as vascular, pulmonary, gastrointestinal, and general surgical procedures [75]. The adhesive has neutral properties and is an excellent agent in stopping bleeding from vessels and from moist surfaces of bleeding parenchymal organs. Therefore, it can be used by all fields of medicine to maintain hemostasis. Researchers are also creating hybrid products such as FloSeal consisting of crosslinked gelatin and thrombin. As with cyanoacrylate adhesives, GRF adhesives are considered potentially cytotoxic and are poorly resorbable. Other GRFG adhesives are still under investigation for their potential use in thoracic and vascular surgery [76].

### 2.4. Chitosan-Based Adhesives

Chitosan is a polysaccharide commercially produced by the deacetylation of chitin, which is a structural element present in the exoskeleton of crustaceans (e.g., crabs and shrimp) and in the cell walls of fungi. The hemostatic properties of chitosan are well known. The patented product Hem Con [77] demonstrated that the adhesives are able to control severe organ bleeding and large venous hemorrhage induced on a pig animal model [78]. The efficacy of the indicated product has been demonstrated for military [79] and civilian use [80]. Powder-based chitosan adhesives have also been developed. One example is Celox, which is a patented formulation in the form of granules prepared from more than one type of chitosan [81,82]. Its efficiency as a hemostatic product was confirmed by Kozen et al. [82] and Kheirabadi et al. [83]. Klokkevolt et al. [84] demonstrated that chitosan effectively reduces oral bleeding from tongue wounds. Lauto et al. [85,86] used a chitosan-based adhesive in vivo in nerve anastomosis in the repair of an injured intestine with good clinical results. This tissue adhesive may be suitable for skin wound closure and in situations requiring urgent hemostasis [87,88]. Chitosan-based adhesives can also effectively close scleral wounds [89]. Hoekstra et al. [90] demonstrated that the use of microcrystalline chitosan (MCCh) as a vascular sheath after arterial catheterization helps restore hemostasis.

### 2.5. Chondroitin Sulfate-Based Adhesives

Chondroitin sulfate is a glycosaminoglycan composed of a chain of sugars (N-acetylgalactosamine and glucuronic acid). The chondroitin chain can contain 100 individual sugars, each of which can be sulfated in different positions and sizes. Chondroitin sulfate is an important structural element of cartilage and provides high resistance to compression [91,92]. Wang et al. [93] described the use of a chondroitin-based adhesive to bind both the patient’s own cartilage tissue and the implanted biomaterial. The developed adhesive integrates the implant with the surrounding own tissues, which is essential for the immediate and long-term recovery of organ function. In vitro and in vivo studies confirmed that the adhesive stimulated extracellular matrix production and tissue regeneration. The adhesive may also be helpful and an alternative to sutures used in ophthalmology during keratoplasty surgery [94].

## 3. Synthetic Adhesives

### 3.1. Cyanoacrylate Adhesives

The history of cyanoacrylates dates back to the 1950s. The bonding properties of cyanoacrylate adhesive were discovered by accident by Eastman Codack Company scientists while working on fairings for jet aircraft. In 1951, while making refractive index measurements, samples of ethyl 2-cyanoacrylate got stuck in the prisms of a refractometer. A few years later, in 1958, the glue first went on sale commercially as Super Glue [95,96,97,98,99,100].

Cyanoacrylate glue was quickly found to exhibit excellent adhesion to a wide variety of materials, including plastics, rubber, metals, ceramics, wood, glass, paper, fabrics, cement, and tissues. [101,102,103]. The monomer cyanoacrylate polymerizes very rapidly in the presence of weak bases without heat or catalyst at room temperature. The rate of polymerization of liquid cyanoacrylate monomer in the presence of weak bases makes it a very fast drying adhesive and cures quickly [104,105,106,107,108]. The chemical structure of cyanoacrylate monomers can be generally characterized as an acrylate backbone with two electronegative groups. The electron-withdrawing groups can undergo anionic polymerization in the presence of bases to produce a glass-like polymer [109,110,111]. Cyanoacrylate adhesives are ideal for medical and dental applications, especially as tissue bonding adhesives and sealing materials.

In dentistry, medicine, and veterinary medicine, agents based on butyl cyanoacrylate, heptyl cyanoacrylate, octyl cyanoacrylate, and ethoxyethyl cyanoacrylate have been used and are distinguished by [102,103,112,113]:-Hydrophilicity, good spreading in living tissue;-Lower heat generation during polymerization, which helps avoid tissue necrosis;-Better mechanical properties (especially elasticity and cosmetic results);-Lesser toxicity—their degradation products can be metabolized and removed from the body.

In the United States, the Food and Drug Administration (FDA) has so far approved only two cyanoacrylate compounds (n-butyl 2-cyanoacrylate and 2-octyl cyanoacrylate), most of which are for external clinical use. In other countries, these adhesives have been approved for internal use. An analysis of 10 years of experience in treating esophageal and gastric variceal bleeding with endoscopic cyanoacrylate adhesive by American scholars from the University of Iowa demonstrated the efficacy in achieving hemostasis in 96% of the treated acute bleeding cases [104]. A study was conducted to evaluate the efficacy and safety of transcatheter embolization with cyanoacrylate adhesive of acute arterial hemorrhages of various etiologies. Analysis of bleeding outcomes showed a 76% therapeutic success rate in patients treated with cyanoacrylate adhesive [105]. The adhesive can also be successfully used for mesh fixation during conventional inguinal hernia repair. Egyptian researchers observed fewer local complications in a study group using cyanoacrylate adhesive and sutures compared to a control group in which the mesh was fixed only with surgical sutures [106]. In dentistry, cyanoacrylate adhesive has been shown to play a beneficial role in promoting soft and hard tissue healing in postextraction wounds during the dental implant placement procedure [107]. Researchers, in comparing the effectiveness of treating dentinal hypersensitivity with laser and cyanoacrylate adhesive, came to some interesting results [108]. In a study conducted on 62 patients with a total of 432 teeth treated, a faster hypersensitivity reduction effect was demonstrated with the adhesive within the first 24 h of treatment. Cyanoacrylate adhesive may be an alternative to sutures after the surgical removal of the third molars. Researchers studied 60 patients who had two molars removed on opposite sides. They found less pain, bleeding, swelling, and sensitivity on the side where cyanoacrylate adhesive was used. Moreover, the number of successfully treated was higher in the adhesive group, 52 vs. 39 [114]. The use of adhesives for skin wound closure has been well described in the literature [10,111,112,113,115,116,117]. One commercially available adhesive is Dermabond. This adhesive is widely used for the closure of small and superficial skin wounds [118]. One disadvantage of cyanoacrylate adhesives is their high price compared to surgical sutures. Glubran2 is a cyanoacrylate adhesive that is approved for use in the European Union in classical and laparoscopic surgery as well as in endoscopy and interventional radiology [119]. It is applied to the skin, provides a good cosmetic result, and eliminates the need for suture removal. A study by Kull et al. [10] showed the better tensile strength of the Glubran2 adhesive on biological substrates compared to a fibrin adhesive. More recent studies also support this. Experts evaluating the appearance of wounds after skin cancer surgery found no statistically significant differences in the aesthetics in wounds closed with cyanoacrylate adhesive compared to surgical sutures [120,121]. The advantages of adhesives are also recognized by plastic surgeons, emergency physicians, and pediatric surgeons who want to protect a child from the trauma of surgical suturing [122]. Adhesives can also be used to treat lower extremity venous insufficiency. A study of 29 patients showed the successful closure of 56 of 57 saphenous veins [123]. There were no complications of deep vein thrombosis or recurrence during the 9-month follow-up period. Dragu et al. [124] demonstrated that a foreign body reaction can occur after the use of an adhesive to treat a superficial wound. While using the adhesive, the risk of wound infection is small and, according to several studies, its values reached the maximum of 8%. [125,126,127].

Adhesives need to be used with the appropriate personal protective equipment in a room with adequate ventilation because cyanoacrylate is a strong adhesive (rapid exothermic polymerization) and exposure to large quantities causes irritation to the eyes, nose, and throat [128]. Additionally, adhesives should only be used away from open flames, sparks, and high temperatures, because both the polymer cyanoacrylate and its monomer sustain combustion [102,129,130].

In the future, the use of cyanoacrylates in medicine will increase as their toxicity decreases and their shelf life is extended for use in external and internal tissues. The second area will be the construction of cyanoacrylate adhesives in the form of nanofibers.

### 3.2. PEG Polyethylene Glycol-Based Adhesives

This group of tissue adhesives consists of linear or branched polyethylene glycol(PEG) molecules that can be crosslinked chemically or under ultraviolet light, depending on the available chemical groups. Photocrosslinking of PEG-based adhesives occurs in the presence of additional photoreactive groups such as acrylate groups [7]. The adhesion mechanism in PEG-based adhesives shows great variability because it is formulation-dependent [131]. FocalSeal-L is an FDA approved adhesive that is crosslinked by irradiation [4,132]. Generally, a PEG-PLA polymer is used, which has a low viscosity. After application to tissue, the PTMC-PEG polymer is added and polymerization is performed by irradiation in the presence of a photoinitiator [133]. This adhesive has been shown to be useful in thoracic surgery to prevent air leakage from suture lines [134]. Application during hemorrhage is difficult and almost impossible due to photoactivation. In an effort to solve this problem, products have been developed that do not require this process. CoSeal consists of two PEG polymers that rapidly crosslink with proteins in tissues and mechanically adhere to synthetic materials [135]. Comparative tests of CoSeal with traditional hemostatic agents such as gelfoam/thrombin for the treatment of anastomotic bleeding have confirmed its efficacy and safety. CoSeal is used in Europe in vascular and cardiac surgery to protect against anastomotic leakage [136,137]. It is indicated for reinforcing sutures and staple lines that may be sources of leakage. Another FDA-approved sealant adhesive is Duraseal, which consists of polyethylene glycol and a solution of trilysine amine. Mixing these two ingredients results in a nontoxic hydrogel with a 3D network. The benefits of the adhesive during sciatic nerve regeneration have also been demonstrated, with treatment results comparable to the traditional method using sutures [138]. SprayGel is a hydrogel that is sprayed onto tissue, where it hydrolyzes after a few days and is then absorbed. This polymer gel has been shown to be safe for use in gynecological procedures and colorectal surgery, including for ileostomy closure [139,140]. AdvaSeal is a PEG-based bioabsorbable photocrosslinker used clinically to seal pulmonary air leakage. Tanaka et al. [141] also demonstrated its efficacy in the treatment of acute aortic dissection.

For PEG-based systems, biodegradation takes 1–6 weeks. The weaknesses of PEG-based adhesive are high swelling after application and its poor adhesion to surrounding tissue. Therefore, one should be careful when using this adhesive in body cavities with limited volume [142].

### 3.3. Urethane-Based Adhesives

Among synthetic materials, urethane-based adhesives are not common. These adhesives are synthesized into prepolymer forms, so that they have the ability to react with the amino groups of proteins that are present in the biological environment and promote adhesion between tissues through the formation of urea bonds. PCL is biodegradable aliphatic polyester that has been approved by the US FDA and has been used as a drug delivery system [143] in the form of absorbable sutures [144] and as a tissue regeneration material [145]. To overcome the drawback of the prolonged curing time of the material, Ferreira et al. [146] presented the modification of polycaprolactone diol (PCL) with isocyanatoethyl 2-methacrylate (IEMA) to form a macromer that will be easily crosslinked by UV irradiation. UV-curable adhesives offer good process control and acceptable cure rates. They have been shown to be well tolerated by various cells and over a wide range of chemical concentrations [147]. Lipatova et al. [72] describes the synthesis of a polyurethane-based adhesive that has already been tested in renal surgery [148] and orthopedic procedures [149], and no toxic effects were observed. Phaneuf and team [150] presented the composition of a new adhesive, consisting of polyurethane based on polyethers containing carboxylic acid groups, which can seal the gaps of vascular prostheses. TissuGlu surgical adhesive is one of the commercially available urethane-based adhesives, and it has been registered for abdominal shell bonding [151]. On the surface of moist tissues, crosslinking occurs spontaneously and forms a strong bond between layers. The adhesive can be degraded by hydrolysis and enzymatic reactions in the presence of lysine. The degradation products are usually polyols (e.g., glycerol), lysine, ethanol, and carbon dioxide, which are water soluble and easily cleared from the body [152]. TissuGlu has been shown to be effective in plastic surgery in preventing seroma formation, which often occurs after abdominoplasty [152]. TissuGlu is nontoxic, absorbable, forms a strong bond between tissue layers, and contributes to the natural healing process.

### 3.4. Biomimetic Adhesives

Scientists are trying to use natural solutions observed in nature to create tissue adhesives. Of particular interest are certain species of snails, worms [153], clams, and geckos. Clams, which are commonly associated as delicacies, have been well studied as a potential source of waterproof adhesive [152,154]. Clams produce and secrete special adhesive substances. These enable them to adhere to substrates in marine environments, which are typically characterized by high salinity, humidity, and turbulence induced by sea waves. Mussel secretions contain proteins and enzymes that have specific adhesive properties, are nontoxic and noninflammatory, and are fully biodegradable [3,155,156]. These proteins are highly abundant in lysine and 3,4-dihydroxyphenyl-L-alanine (DOPA), which contains a catechol functional group [157,158]. The strong adhesion of clams to surfaces is believed to be based on strong covalent and noncovalent binding to surfaces covered by phenolic hydroxyl groups of DOPA or quinone groups [159]. Strong and water-insoluble adhesives from mussels have attracted interest for potential applications in medicine. They can be used as adhesives to bind cells and tissues together. Adhesives are generally environmentally friendly and apparently harmless to the human body and immune system [30,160,161,162]. Despite the identification of the major protein responsible for the adhesive properties and a thorough study of its properties, the practical application of the adhesive based on substances secreted by bivalves is limited. Uneconomical extraction and unsuccessful large-scale production limit the potential use of these adhesives in clinical practice. Obtaining adequate quantities of clams is labor intensive and inefficient [163,164,165]. The chemical extraction process used does not yield pure or individual adhesive proteins. Therefore, the development of clam adhesive has not yet been commercialized.

The ability of gecko to adhere to rough, smooth, vertical, and inverted substrates has led researchers to investigate the morphology and adhesive properties of gecko foot surfaces [166]. Autumn et al. [167] showed that the fibrillar ranks that cover the surface of a gecko’s foot maximize adhesion to surfaces, primarily through adhesive pads that are decorated with a dense set of bristles. Each bristle has multiple spatulas of 200–500 nm in length. The adhesion of the latter to surfaces is controlled by van der Waals forces [168]. Philip Messersmith and his team [169] at Northwestern University in Illinois have combined the techniques by which geckos and bivalves attach to different substrates. In doing so, they created an unusual adhesive substance. The hybrid, bioinspired adhesive consists of an array of nanofibrous polymer fibrils coated with a thin layer of synthetic polymer that mimics wet clam adhesive proteins. The adhesion strength under wet conditions of the nanostructured polymer pillars increased nearly 15-fold when coated with the clam-produced polymer [169]. The system retains its adhesion for over a thousand contact cycles in dry or wet environments. This hybrid adhesive, which combines the characteristic elements of both gecko and clam adhesion, should be useful for reversibly attaching components to various surfaces in any environment. In a study by Mahdavi and Ferreira et. al, [170] a biocompatible and biodegradable gecko-inspired tissue adhesive was developed from a combination of an elastomer and a thin film covering the surface. The adhesive is based on sebacic acid glycerol polyacrylate (PGSA) and a tough biodegradable elastomer with elastic properties. The authors have developed an adhesive with a promising covalent bond to wet tissue that may have applications in medical therapies as a waterproof adhesive to seal anastomoses, adjunctively for the treatment of hernias, ulcers, and burns, and even as a hemostatic agent in wound dressings.

A team of researchers at Harvard University has developed an innovative tissue adhesive modeled after the mucus produced by the snail species *Arion fuscus* [171]. The adhesive exhibits strong adhesion properties to a variety of wet surfaces, including tissue. The adhesive was tested on a beating pig heart, where it was subjected to high pressure. No leakage was demonstrated even when the heart muscle cells were stretched to the maximum. The adhesive can be used as a hemostatic dressing due to its biocompatibility with blood. The adhesive was able to effectively stop acute hemorrhage induced in rat liver. In vitro studies on human fibroblasts showed excellent biocompatibility. The combination of strong adhesion and high deformability could also be used to close skin wounds. Additionally, the adhesive could be used to implant medical devices that support human organs. This adhesive could make a veritable revolution in the way surgical patients are treated in the future [171].

Researchers from China decided to develop an adhesive based on egg white. As one of the most beneficial foods in nature, egg white is inherently nontoxic, biocompatible, and biodegradable. It is abundant in nature, which is associated with the low cost of adhesive manufacturing [172]. The dried protein powder can form a very viscous gel when mixed with a small portion of water. Slides (25 mm × 20 mm) coated with the adhesive can support a weight of 6 kg [172]. During testing, the adhesive at a ratio of 0.875 g of powder per ml of water showed the best injectability from a syringe needle and excellent plasticity and adhesion ability. Because of the irreversible protein aggregation formed during the air-drying process, the adhesion mechanism could be explained by the formation of a hydrogen bonding network and the conformational change of the egg white. A study was conducted on rats using a wound closure adhesive. The effects of treatment were evaluated after 5 days. Normal wound healing without inflammatory reaction or infection was observed [172]. Adhesive from fresh eggs can be produced by air-drying, grinding and mixing with the right amount of water. The resulting adhesive had excellent adhesion strength on various types of substrates and good adhesion under water compared to common commercial medical adhesives. Subcutaneous degradation analysis of the adhesive showed no strong long-term inflammatory response in vivo. Considering its very low production cost and environmentally friendly processing steps, it has a future as a medical adhesive [172].

## 4. Stem Cells in Medical Adhesive Application

Stem cells possess abilities allowing them to differentiate into a plethora of lineage-specific cell types. Despite the primary interest in embryonic stem cells, currently the major focus has been driven towards adult stem cells for the use of tissue repair [173,174]. Adult stem cells are present in several tissues and organs, including the blood vessels, brain, pancreas, or liver [175]. Most commonly, however, mesenchymal stromal cells (MSCs) are derived and gathered from adipose tissue and bone marrow, where they can be collected with use of noninvasive methods [176].

MSCs play a fundamental role in the process of tissue healing, modulating the production of crucial cytokines [177,178,179]. Stem cells’ abilities of promoting angiogenesis and differentiating into various cell types allow them to take an important part in the wound repair process when they are combined with the fibrin matrix [180].

The cutaneous wound healing process is a complex procedure that requires well-coordinated molecular and biological steps in the form of cell migration, angiogenesis, and remodeling [181]. Wu et al. [176], as well as Branski et al. [182], confirmed that both bone marrow stem cells (BMSC) and adipose-derived stem cells (ADSC) are able to enhance the epithelization of cutaneous wounds. Additionally, ADSCs have the potential to induce cellular differentiation towards keratinocytes [183] and enhance the proliferation of fibroblasts [184].

Due to the regenerative properties of MSCs, there have been strategies considering delivering such cells to the site of the injured tissues. According to Ortiz et al. [177], MSCs can be successfully injected directly to the injured site with or without use of biocompatible scaffolds. One of the many examples of stem cell utility in tissue regeneration can be achieved with muscle-derived stem cells, which can enable the regeneration of even critical-sized bone defects [185].

Fibrin adhesive is a noncytotoxic and biocompatible polymer that is naturally biodegradable with the use of the fibrinolytic enzymes of the human body. According to the study [186], fibrin adhesive can be characterized as a highly valuable biomaterial that is capable of delivering vital cells to injured tissues.

Nowadays, tissue engineering is used in attempts to treat a variety of tissue defects that are the result of traumatic injury, but also those that arise during the process of development. Direct injection of cells into the injured tissue gives poor results. A scaffold is usually required to start the proper regeneration process. In the case of regeneration enhancement, fibrin can be used alone or in a combination with specific scaffolds [187]. There are studies confirming the usability of fibrin adhesive as a potential scaffold itself, as well as cell carrier in tissue regeneration [188].

A study [189] confirmed the superiority of using a fibrin gel consisting of autologous rabbit fibrin beads over untreated tibial defects. Another study [190] confirmed the high clinical importance of stem cell-enriched fibrin adhesive in dog bone defects.

The systematic review based on in vivo and in vitro studies confirmed that medical adhesives such as fibrin gel are a suitable carrier and scaffold for stem cell application in case of tissue injuries. Advances in research are still required in order to fully understand the process, yet the whole method has proven to be medically important and promising.

## 5. Discussion

Medical adhesives are becoming more and more popular each year with the studies confirming their utility in regenerative medicine. Generally, suturing in laparoscopic procedures is a most commonly used method for closing the wounds. Yet, the placement of sutures via a laparoscope is a technically challenging and time-consuming method. On the other hand, it has already been confirmed that the use of adhesives in laparoscopic surgeries enables control over bleeding, as well as reinforces the anastomoses. When it comes to the division of adhesives, they can be distinguished according to their origin, natural and synthetic. Among the natural ones, fibrin, together with collagen and gelatin, are one of the most commonly used adhesives.

Fibrin sealants possess features of an excellent template for cellular migrations, supporting keratinocyte and fibroblast growth [191]. Apart from being a hemostatic agent, fibrin sealants can be successfully used in a plethora of clinical situations, including the treatment of gastrointestinal anastomosis leakage, perforated ulcers, or trauma to organs such as the spleen. Research has confirmed that fibrin adhesives can be safely delivered to sites of infection based on animal models.

Collagen sealants have been checked for increased hemostatic activity due to their influence on the platelet adhesion in blood. There are studies [192,193] confirming the increased strength of sealants and adhesives containing collagen fibers.

Gelatin-based adhesives are mostly made of gelatin, which is an irreversibly denatured collagen and a base material for hemostat products. Research confirms that these adhesives can be successfully used in controlling minor bone or vessel bleeding [194].

The second group of adhesives is synthetic. Cyanoacrylate and various other artificial adhesives were invented in order to make up for the weak strength of natural-derived glues. Even though the cyanoacrylate adhesives possess higher strength, potential risks of infections and cytotoxicity were raised [195]. For this reason, the current development of synthetic adhesives focuses on polyethylene glycol or polyurethane-based adhesives.

Nowadays, research concerning adhesives is focused on improving their mechanical properties in order to achieve fully sutureless medical procedures. It is important to underline that most adhesives require applying proper pressure in order to fully stop the hemorrhage. This can be challenging in laparoscopic surgeries due to the limited amount of surgical field and access to the wound. That is why it is important to develop adhesive materials that will become more advanced, with the emphasis on increased mechanical characteristics as well as shortened curing time.

Because of the laparoscopic revolution, the need for enhanced adhesives is even greater than before. Future research should focus on assessing the bioactivity and biocompatibility of new adhesives while maintaining their suitable mechanical characteristics. It should be important to explore the biomechanical properties of native tissues in order for clinicians to better understand the nature and mechanics behind effective adhesives.

## 6. Summary

Over the past several years, thanks to the development of biotechnology, tissue adhesives have changed their composition and structure by modifying existing biomaterials and creating new ones. Some adhesives can be applied topically to the inside of tissues and externally to their surfaces. They are available in various forms (liquid, powder, and sponge). In vitro studies are important for the further development of new materials for in vivo applications. The variety of medical adhesive applications is shown in Table 1. The efficacy of the selected adhesives has already been confirmed, while a large number of new bonding materials await further research. When selecting a tissue adhesive for a specific surgical procedure, cost, side effects, and efficacy must be considered. Some new adhesives show stronger performance compared to those already commercially used. More clinical trials are needed to commercialize new biomaterials. In the future, new tissue adhesives may be designed by mimicking solutions that occur naturally in the environment. The bioengineering of proteins and DNA gene biotechnology has the manufacturing potential to create an adhesive with unique properties. Adhesion strength is one of the most important parameters, so it is important to standardize the tests that tissue adhesives undergo [196]. In addition, more attention should be paid to the matching of the mechanical properties between the adhesive and the target tissue [47]. It is widely believed that adhesives will eventually replace sutures and staplers in surgical procedures, although it is difficult to predict when exactly this will occur.

## Figures and Tables

**Table 1 materials-15-05215-t001:** Types of medical adhesives and their potential application in medicine.

Type of Glue	Application	References
(1) Natural adhesives		
(1a) Fibrin adhesive	hemostasis during cardiac surgery; sealing of vascular grafts; support in the treatment of aortic dissection in vascular surgery; prevention of cerebrospinal fluid leakage during neurosurgery; hemostasis control in bleeding from burn wounds; in endoscopy for the treatment of bleeding from peptic ulcers implantation of biomaterials in inguinal hernia operations; treatment of lymphangiomas; treatment of lymphocele; prevention of lymphatic leakage; use of adhesive for pleurodesis; in thoracic surgery for treatment of bronchial fistulas; for pleural sealing; treatment of damaged nerves; reduction of bleeding in orthopedic surgery	[21,22,23,24,31,32,33,34,35,36,38,39,40,41,42,43,44,45]
(1b) Collagen adhesive	for controlling hemostasis in vascular surgery; the prevention and treatment of cerebrospinal fluid leaks in neurosurgery; the endoscopic treatment of gastrointestinal bleeding	[58,63]
(1c) Gelatin adhesive	cardiac and vascular surgery to produce hemostasis and stop bleeding from vessels; control hemostasis in vascular and organ bleeding in general surgical procedures	[73,75,197]
(1d) Chitosan-based adhesive	staunching of organ and venous bleeding; treatment of damaged nerves, intestines, and closure of skin wounds; treatment of scleral injuries; maintenance of hemostasis after endovascular interventions	[77,78,79,80,82,83,84,85,86]
(1e) Chondroitin-based adhesive	orthopedics for implantation of biomaterials; in ophthalmology as an alternative to sutures	[91,93,94]
(2) Synthetic adhesives		
(2a) Cyanoacrylate-based adhesive	endoscopic treatment of gastrointestinal bleeding; treatment of acute arterial hemorrhage; implantation of biomaterials during inguinal hernia surgery; support of tissue healing during dental implant surgery; treatment of hypersensitivity after dental procedures; closure of skin wounds and mucous membranes; treatment of venous insufficiency of the lower limbs; use for neurological embolization	[104,105,106,107,108,109,110,111,112,113,114]
(2b) Polyethylene glycol (PEG)-based adhesive	prevention of suture line leakage in thoracic surgery and cerebrospinal fluid leakage after neurosurgery; prevention of anastomotic leakage in vascular and cardiac surgery; wound closure and sealing in general surgery, gynecology, and ophthalmology; nerve regeneration.	[131,133,134,136,137,138,139,140]
(2c) Urethane-based adhesive	soft tissue fusion in general, and plastic surgery and orthopedics; sealing of vascular prostheses in angiosurgical procedures.	[72,143,144,145,148]
(3) Biomimetic adhesives		
(3a) Adhesive based on mussel adhesion proteins	potential applications in cell and tissue interfacing	[160,161,162,198]
(3b) Adhesive based on the adhesive properties of the gecko	sealing anastomoses; controlling bleeding from wounds, burns, and ulcers; implantation of biomaterials	[170]
(3c) Adhesive based on snail mucus	control of arterial and parenchymal hemorrhages; skin wound closure; biomaterial implantation	[171]
(3d) Adhesive based on chicken egg white	skin wound closure	[172]

## Data Availability

Not applicable.

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
