# Peer review of "Medical Adhesives and Their Role in Laparoscopic Surgery—A Review of Literature"

_materials, 2022, doi:10.3390/ma15155215_

Round 1
Reviewer 1 Report
This review by Maciej Mazur et al, summarized recent finding and clinical application of natural and synthetic adhesive or glues in medical use. This is a well written review and provide useful information for both clinician and basic researchers. Recommend accept with minor revision.
Few minor comments
- Few recent 5 years literature cited. Maybe add more.
- A recent publication on comparison of fibrin glue and autologous blood clot as scaffold for promoting HMDSCs mediated critical sized calvarial bone defect should be cited in Stem Cell and Medical glue section. (PMID, 34440188).
- Line 41, Missing “4.”
- Line 59-60, the last sentence should be rephrased and emphasize the goal of this review. This sentence not written well.
- Line 77, 78 should specify what “adhesives”
- Line 461, “these” should be “those”.
Author Response
Dear Reviewer,
We would like to express our sincerest gratitude to the Reviewers for their enormous efforts in criticizing the manuscript. All remarks have been included in the revised version of the manuscript.
Reviewer #1
Question 1
Few recent 5 years literature cited. Maybe add more
Answer: We would like to thank you for the comment. New recent literature has been added to the manuscript.
Question 2
A recent publication on comparison of fibrin glue and autologous blood clot as scaffold for promoting HMDSCs mediated critical sized calvarial bone defect should be cited in Stem Cell and Medical glue section. (PMID, 34440188).
Answer: We would like to thank you for the comment. New publication has been added to the manuscript.
Question 3
Line 41, Missing “4.”
Answer: We would like to thank you for the comment, The text has been corrected.
Question 4
Line 59-60, the last sentence should be rephrased and emphasize the goal of this review. This sentence not written well.
Answer: We would like to thank you for the comment, The sentence has been revised.
Question 5
Line 77, 78 should specify what “adhesives”
Answer: We would like to thank you for the comment. The sentence has been revised.
question 6
Line 461, “these” should be “those”.
Answer: We would like to thank you for the comment, The sentence has been revised.
Reviewer 2 Report
The paper subject is relevant for the Journal and writing is very good, but there are major concerns, added to the smaller comments in the attached pdf file:
- Major concern - A review paper supposes to present the latest developments in a given field. In this paper, with 179 references, less than 10% of the references are from the last 5 years (2018- mid 2022), and some of these are not even journal papers nor books. Only 1 and 6 references are from 2022 and 2021, respectively. This is a critical issue for the validity and interest of the work.
- A discussion on future trends and perspectives for each adhesive type is required.
- I would avoid using the term glue, and use always adhesive.
I recommend the authors to address these issues, especially the first one, very carefully, such that the paper could be reconsidered in another review round.

Author Response
Dear Reviewer,
We would like to express our sincerest gratitude to the Reviewers for their enormous efforts in criticizing the manuscript. All remarks have been included in the revised version of the manuscript.
Reviewer #2
Question 1
Major concern - A review paper supposes to present the latest developments in a given field. In this paper, with 179 references, less than 10% of the references are from the last 5 years (2018- mid 2022), and some of these are not even journal papers nor books. Only 1 and 6 references are from 2022 and 2021, respectively. This is a critical issue for the validity and interest of the work.
Answer: We would like to thank you for the comment. The manuscript’s bibliography has been updated.
Question 2
A discussion on future trends and perspectives for each adhesive type is required.
Answer: We would like to thank you for the comment. A discussion section has been added.
Question 3
I would avoid using the term glue, and use always adhesive.
Answer: We would like to thank you for the comment. The text has been revised according to reviewer’s suggestions.
Round 2
Reviewer 2 Report
I noticed that the authors addressed the three main points in my initial report and the paper was improved. Thank you for the improvements. However, I also send attached an annotated PDF file with minor cosmetic issues that could benefit the paper. Maybe the authors didn't see it. I am sending it again for checking. I will recommend to accept after minor revisions so that the authors can correct these issues.

Author Response
We would like to express our sincerest gratitude for your enormous efforts in criticizing the manuscript. The pdf file of the menuscript has been checked, all the necessary corrections have been applied and the authors are sending the final version of the manuscript for acceptance.